# ACTION MATCHING: A VARIATIONAL METHOD FOR LEARNING STOCHASTIC DYNAMICS FROM SAMPLES

## ABSTRACT

Stochastic dynamics are ubiquitous in many fields of science, from the evolution of quantum systems in physics to diffusion-based models in machine learning. Existing methods such as score matching can be used to simulate these physical processes by assuming that the dynamics is a diffusion, which is not always the case. In this work, we propose a method called "Action Matching" that enables us to learn a much broader family of stochastic dynamics. Our method requires access only to samples from different time-steps, makes no explicit assumptions about the underlying dynamics, and can be applied even when samples are uncorrelated (i.e., are not part of a trajectory). Action Matching directly learns an underlying mechanism to move samples in time without modeling the distributions at each time-step. In this work, we showcase how Action Matching can be used for several computer vision tasks such as generative modeling, super-resolution, colorization, and inpainting; and further discuss potential applications in other areas of science.

## 1 INTRODUCTION

The problem of learning stochastic dynamics is one of the most fundamental problems in many different fields of science. In physics, porous medium equations (Vázquez, 2007) describe many natural phenomena from this perspective, such as Fokker Planck equation in statistical mechanics, Vlasov equation for plasma, and Nonlinear heat equation. Another prominent example is from Quantum Mechanics where the state of physical systems is a distribution whose evolution is described by the Schrödinger equation. Recently, stochastic dynamics have achieved very promising results in machine learning applications. The most promising examples of this approach are the diffusion-based generative models (Song et al., 2020b; Ho et al., 2020).

**Informal Problem Setup** In this paper we approach the problem of *Learning Stochastic Dynamics* from their samples. Suppose we observe the time evolution of some random variable $X_t$ with the density $q_t$, from $t_0$ to $t_1$. Having access to samples from the density $q_t$ at different points in time $t \in [t_0, t_1]$, we want to build a model of the dynamics by learning how to move samples in time such that they respect the marginals $q_t$. In this work, we propose a method called "Action Matching" as a solution to this problem.

**Learning Stochastic Dynamics vs. Time-Series** There is an important distinction between the problem of learning stochastic dynamics and time-series modeling (e.g., language, speech or video modeling). In time-series, the samples come in *trajectories*, where the samples in each trajectory are usually highly correlated. However, in learning stochastic dynamics, we only have access to independent samples at any given time-step (i.e., uncorrelated samples through time). This degree of freedom allows us to solve different types of problems that can not be approached by time-series modeling. We provide several examples in our experiment section, but also point out that sometimes it is even physically impossible to obtain samples along trajectories. For example, in Quantum Mechanics, the act of measurement at a given point collapses the wave function which prevents us from obtaining further samples along that trajectory.

**Generative Modeling with Action Matching** From the Machine Learning perspective, the problem of learning stochastic dynamics is a generalization of generative modeling. One way so solve generative modeling is to first construct a distributional path (stochastic dynamics) from the data

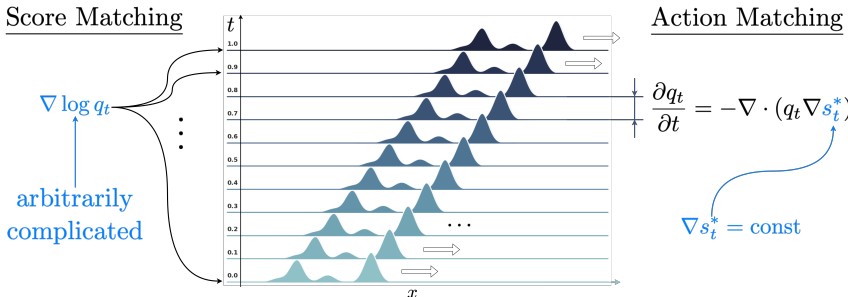

Figure 1: Score Matching learns a model for every distribution, while Action Matching learns the transition rule between distributions according to the continuity equation. Here, we illustrate that learning the dynamics might be a much simpler task than learning all the distributions individually.

distribution to a tractable prior distribution (e.g., Gaussian or uniform), and then learn to move along this path to generate samples. The most prominent example of this approach is the recent developments in diffusion generative models (Song et al., 2020b; Ho et al., 2020), where a stochastic differential equation (SDE) is constructed to move the samples from the data distribution to the prior, and the reverse SDE is constructed by learning the score function of the intermediate distributions via Score Matching (Hyvärinen & Dayan, 2005). Action Matching can be used for generative modeling in a similar way, where we also construct a stochastic dynamics between the data distribution and the prior. However, the important distinction is that this dynamics is constructed solely from *samples* of the intermediate distributions, rather than *analytical SDEs* used in diffusion. This heavily relaxes the constraint on the dynamics required in SDEs, and enables Action Matching to learn a much richer family of dynamics between the two distributions. For example, in both widely used VP-SDEs and VE-SDEs (Song et al., 2020b), the conditionals $q_t(x_t|x_0)$ are tractable Gaussian distributions, while in Action Matching, the dynamics can have any arbitrary conditional $q_t(x_t|x_0)$, as long as it can be sampled from. We can also use Action Matching to learn the dynamics constructed by SDEs as SDEs can be sampled from. In Section 5.1, we provide a rich family of dynamics, that can be learned with Action Matching, without the knowledge of the underlying process.

Another important distinction between SDEs and Action Matching is that the Action Matching modeling capacity is spent only on learning how to move the samples (in a consistent way with the marginals), and does not make any attempt to learn the marginals themselves. However, in diffusion models such as VP-SDEs or VE-SDEs, all the capacity of the model is spent on learning the score function of the individual densities $\nabla \log q_t(x)$ for the backward diffusion. This is wasteful if the evolution of the density is simple, but the densities themselves are complicated. An illustrative toy example of this is provided in Fig. 1, where a complicated density is evolving with a constant velocity through time. In this case, Action Matching only needs to learn a constant velocity vector field, without learning anything about the individual marginals. As a practical example of this, we will consider the colorization task in the experiment section, and argue that moving directly from a grayscale image to the colored image with action matching is much easier than moving from Gaussian noise to a colored image with a conditional diffusion that conditions on the grayscale image.

In short, compared to diffusion generative models, Action Matching has the following advantages:

1. Action Matching relies only on samples and does not require any knowledge of the underlying stochastic dynamics, which is essential when we only have access to samples.

2. Action Matching is designed to learn only the dynamics, rather than the individual distributions $q_t$, which is useful when a complicated distribution has a simple dynamics.

3. Action Matching's applicability extends beyond that of diffusion models, as it can learn a much richer class of stochastic dynamics (see Theorem 1).

Our contribution is two-fold: 1) In Section 2, we discuss a mathematically rigorous problem formulation for learning stochastic dynamics, why this problem is well-defined, and what types of dynamics we aim to learn. 2) In Section 3, we discuss Action Matching as a variational framework for learning these dynamics. Finally, as some of the possible applications of Action Matching, we discuss several

computer vision tasks, such as generative modeling, super-resolution, colorization, and inpainting; and provide experiments.

## 2 PROBLEM FORMULATION OF LEARNING CONTINUOUS DYNAMICS

**Continuity Equation** Suppose we have a set of particles in space $\mathcal{X} \subset \mathbb{R}^d$, initially distributed as $q_{t_0}$. Let each particle follow a time-dependent ODE (continuous flow) with the velocity field $v : [t_0, t_1] \times \mathcal{X} \to \mathbb{R}^d$ as follows

$$\frac{\partial}{\partial t} x(t) = v_t(x(t)), \quad x(t_0) = x. \tag{1}$$

From fluid mechanics, we know that the density of the particles at time $t$, denoted by $q_t$, evolves according to the *continuity equation*

$$\frac{\partial}{\partial t} q_t = -\nabla \cdot (q_t v_t), \tag{2}$$

which holds in the distributional sense, where $\nabla\cdot$ denotes the divergence operator.

Note that even though we arrived at the continuity equation using ODEs, the continuity equation can describe a rich family of density evolutions in a wide range of stochastic processes, including those of SDEs (see Equation 37 of Song et al. (2020b)), or even those of the porous medium equation (Otto, 2001) that are more general than SDEs. This is intuitively because in modeling the density evolution, we only care about respecting the marginals $q_t$, and not the underlying stochastic process that resulted in those marginals. This motivates us to restrict ourselves to ODEs of the form Eq. (1), and the continuity equation, without losing any modelling capacity. In fact, as the following theorem shows, under mild conditions, *any continuous dynamics can be modeled by the continuity equation*, and moreover any continuity equation results in a continuous dynamics.

**Theorem 1** (Adapted from Theorem 8.3.1 of Ambrosio et al. (2008)). *Consider a continuous dynamic with the density evolution of $q_t$, which satisfies mild conditions (absolute continuity in the 2-Wasserstein space of distributions $\mathcal{P}_2(\mathcal{X})$). Then, there exists a unique (up to a constant) function $s_t^*(x)$, called "action", such that vector field $v_t^*(x) = \nabla s_t^*(x)$ and $q_t$ satisfies the continuity equation*

$$\frac{\partial}{\partial t} q_t = -\nabla \cdot (q_t \nabla s_t^*(x)). \tag{3}$$

*In other words, the ODE $\frac{\partial}{\partial t} x(t) = \nabla s_t^*(x)$ can be used to move samples in time such that the marginals are $q_t$.*

Furthermore, we know that $\nabla s_t^*(x)$, defined in Eq. (3), minimizes the *kinetic energy functional* $\mathcal{K}(v_t)$, defined below, along $q_t$ (Ambrosio et al., 2008)

$$\mathcal{K}(v_t) := \frac{1}{2} \int_{t_0}^{t_1} \mathbb{E}_{q_t(x)} \|v_t(x)\|^2 dt, \qquad \nabla s_t^* = \arg\min_{v_t} \left\{ \mathcal{K}(v_t) \,\middle|\, \frac{\partial}{\partial t} q_t = -\nabla \cdot (q_t v_t) \right\}, \tag{4}$$

where the optimization is over all $v_t$ satisfying the continuity equation with $q_t$. We can use the optimal value of the optimization in Eq. (4) to attribute a unique *kinetic energy* value $\mathcal{K}$ to any stochastic dynamics $q_t$ as follows:

$$\mathcal{K} := \frac{1}{2} \int_{t_0}^{t_1} \mathbb{E}_{q_t(x)} \|\nabla s_t^*(x)\|^2 dt, \tag{5}$$

Using Theorem 1, the problem of learning the dynamics can be boiled down to learning the unique vector field $\nabla s_t^*$, only using samples from $q_t$. Motivated by this, we restrict our search space of velocity vectors to the family of curl-free vector fields

$$\mathcal{S}_t = \{\nabla s_t \,|\, s_t : \mathcal{X} \to \mathbb{R}\}. \tag{6}$$

We use a neural network to parameterize the set of functions $s_t(x)$, and propose Action Matching for learning the neural network such that $s_t(x)$ approximates $s_t^*(x)$. Once we have learned the vector field $\nabla s_t^*$, we can move samples forward or backward in time by simulating the ODE in Eq. (1) with the velocity $\nabla s_t^*$. The continuity equation ensures that samples at any given time $t \in [t_0, t_1]$ are distributed according to $q_t$.

# 3 ACTION MATCHING

The main development of this paper is the Action Matching method, which allows us to recover the *true action $s_t^*$* of a continuous dynamic and thereby simulate it, while having access only to samples from $q_t$. In order to do so, we define the *variational action $s_t(x)$*, parameterized by a neural network, that approximates $s_t^*(x)$, by minimizing the "ACTION-GAP" objective

$$\text{ACTION-GAP}(s, s^*) \coloneqq \frac{1}{2} \int \mathbb{E}_{q_t(x)} \| \nabla s_t(x) - \nabla s_t^*(x) \|^2 dt \,. \tag{7}$$

Note that this objective is intractable, as we do not have access to $\nabla s^*$. We now propose action matching as a variational framework for optimizing this objective.

We first show that the problem of minimizing the intractable Eq. (7), is tightly related to estimating another intractable quantity: the kinetic energy of a continuous dynamics. As discussed in Theorem 1, we can attribute a kinetic energy $\mathcal{K}$ quantity to any absolutely continuous dynamics $q_t$, using the true action $s^*$ (see Eq. (4)). In order to estimate the intractable $\mathcal{K}$, we define a tractable variational *kinetic energy lower bound* (KILBO) as a functional of an arbitrary variational action $s$ as follows:

$$\text{KILBO}(s) = \underbrace{\mathbb{E}_{q_{t_1}(x)}[s_{t_1}(x)] - \mathbb{E}_{q_{t_0}(x)}[s_{t_0}(x)]}_{\text{action-increment}} - \underbrace{\int \mathbb{E}_{q_t(x)} \left[ \frac{1}{2} \| \nabla s_t(x) \|^2 + \frac{\partial s_t}{\partial t}(x) \right] dt}_{\text{smoothness (regularization)}} \,. \tag{8}$$

The following proposition establishes that Eq. (7) is the gap between KILBO and the true kinetic energy.

**Proposition 1.** *For an arbitrary variational action $s$, KILBO$(s)$ is a lower bound on the true kinetic $\mathcal{K}$, and the gap can be characterized with*

$$\mathcal{K} = \text{KILBO}(s) + \text{ACTION-GAP}(s, s^*) \,. \tag{9}$$

*Thus, since $\mathcal{K}$ is not a function of $s$, the following optimization problems are equivalent*

$$\arg\max_s \{\text{KILBO}(s)\} = \arg\min_s \{\text{ACTION-GAP}(s, s^*)\} \,, \tag{10}$$

*where the equality is up to an additive constant. The KILBO gap is tight iff $\nabla s_t(x) = \nabla s_t^*(x)$.*

See Appendix A for the proof. Proposition 1 indicates that maximizing the KILBO results in estimating the true kinetic energy, as well as matching the variational action to the true action. Note that unlike the intractable $\mathcal{K}$, maximizing KILBO is tractable, as we can use the samples of $q_t$ to obtain an unbiased low variance estimate of KILBO.

KILBO can be decomposed into an *action-increment* and a smoothness term. If we only optimize the action-increment term, we learn large values for $s_t(x)$ at $t_1$ and small values for $s_t(x)$ at $t_0$. In this case, $s_t(x)$ tends to learn a degenerate function with sharp transitions in both $x$ and $t$ directions. The smoothness term acts as a regularization term by penalizing large gradients with respect to both $x$ direction $\left( \frac{1}{2} \| \nabla s_t(x) \|^2 \right)$, and $t$ direction $\left( \frac{\partial s_t(x)}{\partial t} \right)$.

# 4 GENERATIVE MODELING USING ACTION MATCHING

While Action Matching has a wide range of applications in learning the continuous dynamics, in this work, we focus on the applications of Action Matching in generative modeling task. In Action Matching generative models, we first have to define a dynamics (i.e., noising process) that transforms samples from the data distribution $q_0 = \pi$ to samples of a prior distribution $q_1$ (e.g., standard Gaussian). Action Matching is then used to learn the vector field $\nabla s^\star$ of the chosen dynamics. Once $\nabla s^\star$ is learned, we can sample from the target distribution by first sampling from the prior, and then moving the samples using a reverse ODE with the velocity $\nabla s^\star$. Finally, Action Matching enables use to compute the exact log-likelihood of the data.

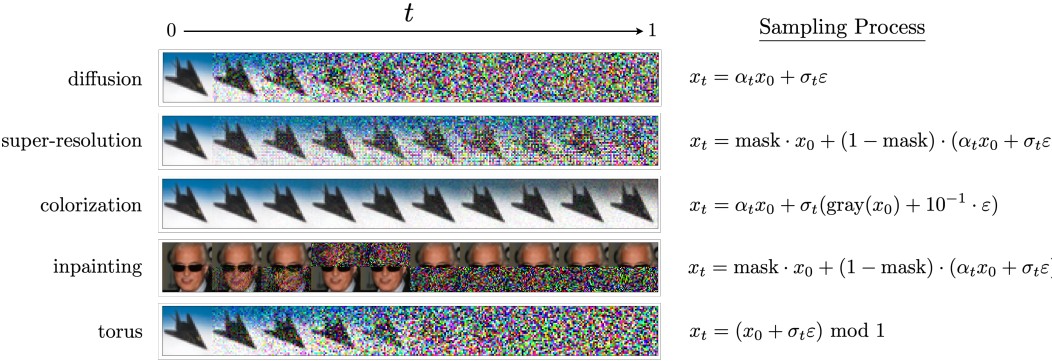

Figure 2: Examples of different noising processes used for different vision tasks. At $t = 0$, we start from the data distribution. Depending on the task, the noising process gradually destroys all or partial information of data, and replace it with prior noise.

## 4.1 NOISING PROCESSES IN ACTION MATCHING GENERATIVE MODELS

To learn the vector field $\nabla s^\star$, Action Matching only requires samples from the intermediate distributions $q_t, t \in [0, 1]$, that define the noising process. We now provide a broad family of noising processes that can be used for generative modeling tasks. Consider the process

$$x_t = f_t(x_0) + \sigma_t \varepsilon, \quad x_0 \sim \pi(x), \quad \varepsilon \sim p(\varepsilon), \tag{11}$$

where $f_t(x_0)$ is some transformation of the data, which could be nonlinear. At $t = 0$, $f_0$ is the identity function, and $\sigma_0 = 0$. Thus, $x_0$ is distributed according to the data distribution, i.e., $q_0(x_0) = \pi(x_0)$. The noising process then gradually eliminates information from the samples using $f_t$, and increases the variance of noise $\sigma_t$. At $t = 1$, $f_t$ would become the zero function and we have $\sigma_1 = 1$. Thus, $x_1$ would be distributed as $q_1(x_1) = p(x_1)$.

We now demonstrate that the same general idea can be used to construct different noising processes for solving different vision tasks, such as diffusion image generation, super-resolution, colorization, inpainting, and torus image generation. See Fig. 2 for the examples of these sampling processes. We will demonstrate Action Matching learning these dynamics in the experiment section.

**Action Matching for Learning the Diffusion Dynamics**    Diffusion processes can be viewed as a special case of the process in Eq. (11), when $f_t(x_0)$ is a linear transformation $f_t(x_0) = \alpha_t x_0$

$$x_t = \alpha_t x_0 + \sigma_t \varepsilon, \quad \varepsilon \sim \mathcal{N}(0, 1), \tag{12}$$

where $\alpha_t$ and $\sigma_t$ we can be chosen such that the marginals of Eq. (12) corresponds to the marginals of VP-SDE and VE-SDE (Equation 29 from Song et al. (2020b)). This sampling process corresponds to the unconditional image generation task since this dynamics transforms all the information in the image into the Gaussian noise. We can use Action Matching to learn the dynamics of Eq. (12), solely from samples of Eq. (12), *without any knowledge of the underlying diffusion process*.

The Equations (11) and (12) showcase that Action Matching generalizes diffusion, by allowing $f_t$ to be any non-linear function, $\varepsilon \sim p$ be any noise model. In contrast, Denoising Score Matching uses linear function $f_t$ for VP-SDE and VE-SDE, with the tractable Gaussian conditional.

**Super-Resolution and Inpainting**    Action Matching provides a lot of freedom in the choice of the sampling process, which we demonstrate in the conditional generation tasks. Consider the following sampling processes

$$x_t = \text{mask} \cdot x_0 + (1 - \text{mask}) \cdot (\alpha_t x_0 + \sigma_t \varepsilon), \tag{13}$$

where the `mask` variable has the same dimensions as $x_0$ and every coordinate of the mask vector is in $\{0, 1\}$. Thus, the noising process is only applied to the subset of pixels, which can be used to learn the inpainting and super-resolution tasks. In the inpainting task, the mask itself is a Bernoulli random variable that decides either the top-half or the bottom-half of the image is destroyed. In the super-resolution case, the mask is fixed, and keeps one pixel in each $2 \times 2$ block, while the remaining pixels are transformed to noise.

---

**Algorithm 1** Generative Modeling using Action Matching

---

**Require:** dataset $\{x^i\}_{i=1}^N$, $x^i \sim \pi(x) = q_0(x)$
**Require:** parametric model $s_t(x, \theta)$
  **for** learning iterations **do**
    sample a batch of data $\{x_0^i\}_i^n \sim \pi(x) = q_0(x_0)$
    sample a batch of noise $\{\varepsilon^i\}_i^n \sim q_1(x_1) = p(\varepsilon)$
    sample times $\{t^i\}_i^n \sim \text{Uniform}[0, 1]$
    sample two batches $\{x_1^i\}_i^n, \{x_{t^i}^i\}_i^n$ using $x_{t^i}^i = f_{t^i}(x_0^i) + \sigma_{t^i}\varepsilon^i$
    $\text{L} = \frac{1}{n}\sum_i^n \left[ s_0(x_0^i) - s_1(x_1^i) + \frac{1}{2}\left\| \nabla s_t(x_{t^i}^i) \right\|^2 + \frac{\partial s_t(x_{t^i}^i)}{\partial t} \right]$
    update the model $\theta \leftarrow \text{Optimizer}(\theta, \nabla_\theta \text{L}_\theta)$
  **end for**
  **return** trained model $s_t(x, \theta^*)$

---

**Colorization** Another option for the conditional generation is the interpolation between the original datapoint and its nonlinearly transformed version with some noise added. For instance, we do image colorization using this approach

$$x_t = \alpha_t x_0 + \sigma_t (10^{-1}\varepsilon + \text{gray}(x_0)), \tag{14}$$

where the function $\text{gray}(x_0)$ returns the grayscale version of image $x_0$. Note that function $\text{gray}(x_0)$ is not injective, since it maps several different colorizations to the same grayscale image. As we show further, Action Matching sampling process is a bijection; hence, the addition of noise is crucial for sampling from the data distribution images given the same conditioning. At the same time, the added noise partially destroys the information. To avoid this corruption of the conditional image, we concatenate the original grayscale image with the input.

**Generative Modeling on a Torus** Finally, we consider the problem of learning a stochastic dynamics on a manifold. Here we consider a distribution on torus, where every coordinate of the data vector to be in $[0, 1]$ with periodic boundary conditions. Then, the sampling process interpolating between the data distribution on the noise distribution is

$$x_t = (x_0 + \sigma_t \varepsilon) \mod 1, \quad \varepsilon \sim \mathcal{N}(0, 1). \tag{15}$$

Note that $q_1$ converges to the uniform distribution on the torus when $\sigma_t \to \infty$.

## 4.2 Learning, Sampling, and Likelihood Evaluation of Action Matching Generative Models

**Learning** Once we define the noising process for $q_t$, $\forall t \in [0, 1]$, we apply Action Matching as described in Algorithm 1. It samples points with different time-steps and then minimizes the objective (7) w.r.t. the parameters $\theta$ of $s_t(x, \theta)$. In practice, we found that the performance of Algorithm 1 might be hindered by high variance of the objective estimate. To reduce the variance of the objective (7), we propose to weight it over time and also adaptively select the distribution of sampled time-steps. We derive the *weighted* KILBO objective in Appendix A, and further discuss the details of training in Appendix B.

**Sampling** We sample from the target distribution via the trained function $s_t(x(t), \theta^*)$ by solving the following ODE backward in time:

$$\frac{\partial}{\partial t}x = \nabla_x s_t(x(t), \theta^*), \quad x(t = 1) = \varepsilon, \quad \varepsilon \sim p(\varepsilon). \tag{16}$$

Recall that this sampling process is justified by Eq. (3), where $s_t(x(t), \theta^*)$ approximates $s_t^*$.

**Evaluating the Log-likelihood** for the generation tasks can be done by integrating the same ODE forward, i.e.,

$$\log q_0(x(0)) = \log q_1(x(1)) + \int_0^1 dt \, \nabla^2 s_t^*(x(t)), \quad \frac{\partial}{\partial t}x = \nabla_x s_t^*(x(t)), \quad x(t = 0) = x, \tag{17}$$

where we approximate $s_t^*$ by $s_t(x(t), \theta^*)$ and assume the density $q_1(x)$ to be a known analytic distribution.

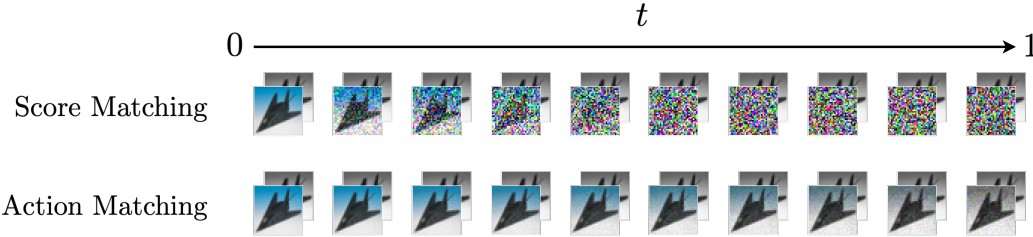

Figure 3: Illustration of the difference between Score Matching and Action Matching noising processes on the colorization task. We argue that Action Matching provides a more efficient way to learn the colorization model since the process requires less changes between the input and the resulting images. The additional channels are used to condition all the inputs on the grayscale image.

### 4.3 ACTION MATCHING VS. SCORE MATCHING GENERATIVE MODELS

In this section, we give more insights on Action Matching by drawing connections and highlighting differences to Score Matching (Hyvärinen & Dayan, 2005) and recently introduced generative models relying on Score Matching (Song et al., 2020b; Ho et al., 2020).

The most fundamental difference between Action Matching and Score Matching is that they approach completely different estimation problems. Indeed, Score Matching estimates the gradient of the log-density from samples of the distribution, while Action Matching cannot be applied for a single distribution. Instead, Action Matching learns the underlying mechanism of a stochastic dynamics, i.e., learns how distributions change in time. We schematically depict this difference in Fig. 1.

Despite the fundamental difference in the problem setup both methods can be applied for generative modeling. For generative modeling, we have the freedom to choose the stochastic dynamics between the target distribution and the prior. Hence, by choosing the dynamics to be a diffusion (Fokker-Planck equation) with known drift and diffusion coefficients, one can learn the score of every marginal and then sample using the corresponding ODE or SDE (Song & Ermon, 2019). Furthermore, if the drift term is affine then Denoising Score Matching (Vincent, 2011) can be applied to learn the score model. More precisely, the model from (Song et al., 2020b) requires samples from the noising process and the analytic formulas for the drift and diffusion coefficient, where the drift term has to be affine.

Action Matching requires only samples from the process to learn the dynamics. Hence, it includes the case of diffusion even without the knowledge of the drift and diffusion coefficient. Moreover, it can learn a much broader family of generative processes, which can have better properties for different applications. In Fig. 3, we give an example of such a process for the colorization task. Since Score Matching can be defined only for the diffusion, its forward process removes all the information about the image resulting in pure noise. In contrast, for Action Matching, we can remove just the information about the color of image while adding some low-variance noise along the way. For both models, we concatenate the grayscale image with the input. However, we argue that, for Action Matching, less computational efforts are needed since we have to apply less modifications to the original image to color it. We discuss and provide evidence for this in Section 5.1.

## 5 EXPERIMENTS

### 5.1 GENERATIVE MODELING

Action Matching has a wide range of applications in modeling density evolutions. In this section, we showcase the applications of Action Matching in generative modeling tasks, since they are among the most challenging high-dimensional stochastic dynamics. Action Matching generative models should not be directly compared with diffusion models, as they make different assumptions, and have access to different information. A score matching diffusion model such as VE-SDE and VP-SDE explicitly relies on the analytic forms of drift and the diffusion coefficients of the SDE. In contrast, Action Matching infers the underlying vector field of any arbitrary continuous stochastic dynamics, solely from the samples. For this reason, we expect Action Matching generative models to under-perform in this setting.

Table 1: Experimental results for Action Matching (AM) and Score Matching (SM) on computer vision tasks. Diffusion and Torus map images to known distributions; hence, for them, we report negative log-likelihood in bits per dimension (BPD). For all tasks, we report FID evaluated between generated images and the test data. For CelebA, we use 20k images. For CIFAR-10, we use 10k images.

| Dataset | Task | BPD↓ | | FID↓ | |
|---|---|---|---|---|---|
| | | SM | AM | SM | AM |
| CelebA | Diffusion | 2.56 | 3.78 | 4.60 | 18.07 |
| CelebA | Superres | – | – | 1.22 | 4.92 |
| CelebA | Inpainting | – | – | 2.02 | 10.71 |
| CelebA | Torus | – | 3.90 | – | 18.09 |
| CIFAR-10 | Diffusion | 3.19 | 4.31 | 12.05 | 53.86 |
| CIFAR-10 | Superres | – | – | 5.94 | 26.42 |
| CIFAR-10 | Colorization | – | – | 5.35 | 7.91 |
| CIFAR-10 | Torus | – | 6.42 | – | 39.42 |

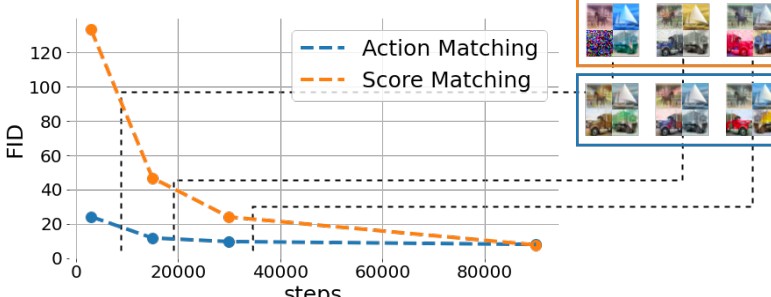

Figure 4: Faster convergence of Action Matching (AM) compared to Score Matching (SM) in FID values and generated samples quality for the colorization task on CIFAR-10.

We apply Action Matching to MNIST, CelebA (Liu et al., 2015) and CIFAR-10 datasets for a variety of computer vision tasks. Namely, we perform unconditional image generation via diffusion as well as conditional generation for super-resolution, in-painting, and colorization tasks. In addition to these settings, we also learn unconditional image generation on a torus, where Denoising Score Matching can not be applied in the original formulation.

For the baseline in unconditional image generation tasks, we use the model from (Song et al., 2020b), which is the diffusion-based generative process trained with Denoising Score Matching. For the baseline in conditional image generation tasks, we follow (Saharia et al., 2022), and condition the model by concatenating the conditioning image as an additional channel with the main input. We refer to all baselines as Score Matching (SM in Table 1 and Fig. 4). We discuss further implementation details in Appendix D.1.

We train all models for 300k iterations and report the negative log-likelihood in bits per dimension (BPD) and FID scores (Heusel et al., 2017) in Table 1. We demonstrate generated images by Action Matching in Appendix E and provide animations of the generation in github.com/action-matching.

We observe that Denoising Score Matching performs better than Action Matching on all tasks, which was expected due to the additional information that the Denoising Score Matching objective uses about the underlying process. However, as we discussed in Section 4.3, we expect Action Matching to converge faster on the conditional image generation tasks, as it only needs to learn a cross-domain transformation, rather than learning the conditional generation from the Gaussian noise. We experimentally verified this hypothesis by evaluating the FID throughout the training process, on the colorization task, shown in Fig. 4.

## 5.2 SCHRÖDINGER EQUATION SIMULATION

In this section, we demonstrate that Action Matching can learn a wide range of stochastic dynamics by applying it to the dynamics of a quantum system evolving according to the Schrödinger equation. The Schrödinger equation describes the evolution of many quantum systems, and in particular, it describes the physics of molecular systems. Here, for the ground truth dynamics, we take the dynamics of an excited state of the hydrogen atom, which is described by the following equation

$$i\frac{\partial}{\partial t}\psi(x,t) = -\frac{1}{\|x\|}\psi(x,t) - \frac{1}{2}\nabla^2\psi(x,t). \tag{18}$$

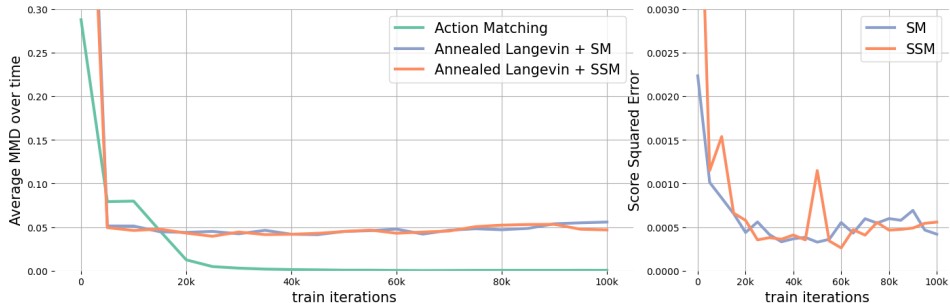

Figure 5: On the left, we demonstrate performance of compared algorithms in terms of average MMD over the time of dynamics. The MMD is measured between generated samples and the training data. On the right, we report squared error of the score estimation for the score-based methods.

| Method | Average MMD↓ |
|---|---|
| AM (ours) | $5.7 \cdot 10^{-4} \pm 3.1 \cdot 10^{-4}$ |
| ALD + SM | $4.8 \cdot 10^{-2} \pm 4.8 \cdot 10^{-3}$ |
| ALD + SSM | $4.7 \cdot 10^{-2} \pm 4.0 \cdot 10^{-3}$ |
| ALD + True Scores | $3.6 \cdot 10^{-2} \pm 4.1 \cdot 10^{-4}$ |

Table 2: Performance of Action Matching and the Annealed Langevin Dynamics (ALD) for the Schrödinger equation simulation. For ALD, we estimate the scores in two ways: Score Matching (SM) and Sliced Score Matching (SSM). We also demonstrate that even using true scores does not allow for the precise simulation.

The function $\psi(x, t) : \mathbb{R}^3 \times \mathbb{R} \to \mathbb{C}$ is called a wavefunction and it completely describes the state of the quantum system. In particular, it defines the distribution of the coordinates $x$ by defining its density as $q_t(x) := |\psi(x, t)|^2$, which dynamics is defined by the dynamics of $\psi(x, t)$ in Eq. (18).

For the baseline, we take Annealed Langevin Dynamics as considered in (Song & Ermon, 2019). It approximates the ground truth dynamics using only scores of the distributions by running the approximate MCMC method (which does not have access to the densities) targeting the intermediate distributions of the dynamics (see Algorithm 3). For the estimation of scores, we consider Score Matching (SM) (Hyvärinen & Dayan, 2005), Sliced Score Matching (SSM) (Song et al., 2020a), and additionally evaluate the baseline using the ground truth scores. For further details, we refer the reader to Appendix D.2 and the code github.com/action-matching.

Action Matching outperforms both Score Matching and Sliced Score Matching, precisely simulating the true dynamics (see Fig. 5 and Table 2). Despite that both SM and SSM accurately recover the ground truth scores for the marginal distributions (see the right plot in Fig. 5), one cannot efficiently use them for the sampling from ground truth dynamics. Note, that even using the ground truth scores in Annealed Langevin Dynamics does not match the performance of Action Matching (see Table 2) since it is itself an approximation to the Metropolis-Adjusted Langevin Algorithm. Finally, we provide animations of the learned dynamics for different methods (see github.com/action-matching) to illustrate the performance difference.

# 6 CONCLUSION

In this work, we discussed how any continuous dynamics (under mild conditions) can be represented by a unique continuous vector field minimizing the kinetic energy. This representation provides a rigorous mathematical formulation for the problem of learning stochastic dynamics. We then presented Action Matching, as a variational framework for learning this unique vector field solely from samples of the dynamics. We further demonstrated that Action Matching can learn a wide range of continuous dynamics, including those of diffusion. We believe the flexibility that Action Matching introduces will be useful in applications in natural sciences, where stochastic dynamics appear, but the underlying mechanisms are not controlled, and thus we can only make observations.

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

## A   ACTION MATCHING

**Proposition.** *For an arbitrary variational action $s$,* KILBO$(s)$ *is a lower bound on the true kinetic* $\mathcal{K}(s^*)$, *and the gap can be characterize with*

$$\text{KILBO}(s) = \omega_{t_1}\mathbb{E}_{q_{t_1}(x)}[s_{t_1}(x)] - \omega_{t_0}\mathbb{E}_{q_{t_0}(x)}[s_{t_0}(x)] - \int_{t_0}^{t_1}\omega_t\mathbb{E}_{q_t(x)}\left[\frac{1}{2}\|\nabla s_t(x)\|^2 + \frac{\partial s_t(x)}{\partial t} + s_t(x)\frac{d\log\omega_t}{dt}\right]dt$$

$$= \mathcal{K} - \text{ACTION-GAP}(s, s^*),$$

*Thus, since $\mathcal{K}$ is not a function of $s$, the following optimization problems are equivalent*

$$\arg\max_s\{\text{KILBO}(s)\} = \arg\min_s\{\text{ACTION-GAP}(s, s^*)\}, \tag{19}$$

*where the equality is up to an additive constant. The* KILBO *gap is tight iff $\nabla s_t(x) = \nabla s_t^*(x)$.*

*Proof.*

ACTION-GAP$(s, s^*)$

$$= \frac{1}{2}\int_{t_0}^{t_1}\omega_t\mathbb{E}_{q_t(x)}\|\nabla s - \nabla s^*\|^2 dt$$

$$= \frac{1}{2}\int_{t_0}^{t_1}\int_x\omega_t q_t(x)\|\nabla s - \nabla s^*\|^2 dxdt$$

$$= \frac{1}{2}\int_{t_0}^{t_1}\int_x\omega_t q_t(x)\|\nabla s\|^2 dxdt - \int_{t_0}^{t_1}\omega_t\int_x q_t(x)\langle\nabla s_t(x), \nabla s_t^*(x)\rangle dxdt + \overbrace{\frac{1}{2}\int\mathbb{E}_{q_t(x)}\|\nabla s^*\|^2 dt}^{\mathcal{K}}$$

$$= \frac{1}{2}\int_{t_0}^{t_1}\int_x\omega_t q_t(x)\|\nabla s\|^2 dxdt - \int_{t_0}^{t_1}\omega_t\int_x\langle\nabla s_t(x), q_t(x)\nabla s_t^*(x)\rangle dxdt + \mathcal{K}$$

$$\overset{(1)}{=} \frac{1}{2}\int_{t_0}^{t_1}\int_x\omega_t q_t(x)\|\nabla s\|^2 dxdt + \int_{t_0}^{t_1}\omega_t\int_x s_t(x)[\nabla\cdot(q_t(x)\nabla s_t^*(x))]dxdt + \mathcal{K}$$

$$= \frac{1}{2}\int_{t_0}^{t_1}\int_x\omega_t q_t(x)\|\nabla s\|^2 dxdt - \int_{t_0}^{t_1}\left(\int_x\omega_t s_t(x)\frac{\partial q_t(x)}{\partial t}dx\right)dt + \mathcal{K}$$

$$\overset{(2)}{=} \int_{t_0}^{t_1}\omega_t\mathbb{E}_{q_t(x)}\left[\frac{1}{2}\|\nabla s_t(x)\|^2\right]dt - \left(\omega_t\mathbb{E}_{q_t(x)}[s_t(x)]\big|_{t_0}^{t_1} - \int_x\mathbb{E}_{q_t(x)}\left[s_t(x)\frac{d\omega_t}{dt} + \omega_t\frac{\partial s_t(x)}{\partial t}\right]dt\right) + \mathcal{K}$$

$$= \int_{t_0}^{t_1}\omega_t\mathbb{E}_{q_t(x)}\left[\frac{1}{2}\|\nabla s_t(x)\|^2 + \frac{\partial s_t(x)}{\partial t} + s_t(x)\frac{d\log\omega_t}{dt}\right]dt - \omega_{t_1}\mathbb{E}_{q_{t_1}(x)}[s_{t_1}(x)] + \omega_{t_0}\mathbb{E}_{q_{t_0}(x)}[s_{t_0}(x)] + \mathcal{K}$$

$$= -\text{KILBO}(s) + \mathcal{K},$$

where in (1), we have used $\int_V\langle\nabla g, \boldsymbol{f}\rangle dx = \oint_{\partial V}\langle\boldsymbol{f}g, \boldsymbol{ds}\rangle - \int_V g(\nabla\cdot\boldsymbol{f})dx$, and in (2) we have used integration by parts. □

## B   GENERATIVE MODELING IN PRACTICE

In practice, we found that the naive application of Action Matching (Algorithm 1) for complicated dynamics such as image generation might exhibit poor convergence due to the large variance of objective estimate. Moreover, the optimization problem

$$\min_{s_t}\frac{1}{2}\int q_t^*(x)\|\nabla s_t(x) - \nabla s_t^*(x)\|^2 dxdt \tag{20}$$

might be ill posed due to the singularity of the ground truth vector field $\nabla s_t^*$. Indeed, consider the sampling process

$$x_t = f_t(x_0) + \sigma_t\varepsilon, \ \ x_0 \sim \pi(x), \ \ \varepsilon \sim \mathcal{N}(x\,|\,0, 1), \tag{21}$$

where the target distribution is a mixture of delta-functions

$$\pi(x) = \frac{1}{N}\sum_i^N\delta(x - x^i). \tag{22}$$

---

**Algorithm 2** Generative Modeling using Action Matching (In Practice)

---

**Require:** dataset $\{x^i\}_{i=1}^N$, $x^i \sim \pi(x) = q_0(x)$
**Require:** parameteric model $s_t(x, \theta)$, weight schedule $\omega(t)$
  **for** learning iterations **do**
    sample a batch of data $\{x_0^i\}_i^n \sim \pi(x) = q_0(x)$
    sample a batch of noise $\{\varepsilon^i\}_i^n \sim q_1(x_1)$
    sample times $\{t^i\}_i^n \sim p(t)$
    sample two batches $\{x_1^i\}_i^n$, $\{x_{t^i}^i\}_i^n$ using $x_{t^i}^i = f_{t^i}(x_0^i) + \sigma_{t^i}\varepsilon^i$
    $\mathrm{L} = \sum_i^n \frac{1}{p(t^i)} \left[ s_0(x_0^i)\omega(0) - s_1(x_1^i)\omega(1) + \frac{1}{2}\left\|\nabla s_t(x_{t^i}^i)\right\|^2 \omega(t^i) + \frac{\partial s_t(x_{t^i}^i)}{\partial t}\omega(t^i) + s_t(x_t^i)\frac{\partial \omega(t^i)}{\partial t^i} \right]$
    update the model $\theta \leftarrow \mathrm{Optimizer}(\theta, \nabla_\theta \mathrm{L}_\theta)$
  **end for**
  **return** trained model $s_t(x, \theta^*)$

---

Denoting the distribution of $x_t$ as $q_t(x)$, we can solve the continuity equation

$$\frac{\partial q_t}{\partial t} = -\nabla \cdot (q_t \nabla s_t^*) \tag{23}$$

analytically (see Appendix C). The ground truth vector field is

$$\nabla s_t^* = \frac{1}{\sum_i q_t^i} \sum_i q_t^i \left[ (x - f_t(x^i))\frac{\partial}{\partial t}\log\sigma_t + \frac{\partial f_t(x^i)}{\partial t} \right], \quad q_t^i(x) = \mathcal{N}(x \mid f_t(x^i), \sigma_t^2). \tag{24}$$

For generative modeling, it's essential that $q_0 = \pi(x)$; hence, $\lim_{t\to 0}\sigma_t = 0$ and $\lim_{t\to 0} f_t(x) = x$. Assuming that $\sigma_t^2$ is continuous and differentiable at 0, in the limit, we have

$$\lim_{t\to 0}\|\nabla s_t^*(x)\|^2 \propto \lim_{t\to 0}\frac{1}{\sigma_t^2}, \quad \text{and} \quad \lim_{t\to 0}\frac{1}{2}\int q_t^*(x)\|\nabla s_t^*(x)\|^2 dx dt \propto \lim_{t\to 0}\frac{1}{\sigma_t^2}. \tag{25}$$

Thus, the loss can be properly defined only on the interval $t \in (\delta, 1]$, where $\delta > 0$. In practice, we want to set $\delta$ as small as possible; hence, ideally, we want to learn $s_t$ on the whole interval $t \in [0, 1]$. We can get rid of the singularity just by reweighting the objective in time, i.e.,

$$\frac{1}{2}\int q_t^*(x)\|\nabla s_t(x) - \nabla s_t^*(x)\|^2 dx dt \implies \frac{1}{2}\int \omega(t)q_t^*(x)\|\nabla s_t(x) - \nabla s_t^*(x)\|^2 dx dt. \tag{26}$$

To give an example, we can take $\sigma_t = \sqrt{t}$ and $f_t(x) = x\sqrt{1-t}$, then $\omega(t) = (1-t)t^{3/2}$ cancels out the singularities at $t = 0$ and $t = 1$.

The second modification of the original Algorithm 1 is the sampling of time-steps for the estimation of the time integral. Namely, the optimization of (26) is equivalent to the minimization of the following objective

$$\mathrm{L}(s) = \underbrace{\int \omega(t_1)s_{t_1}(x)q_{t_1}^*(x)dx - \int \omega(t_0)s_{t_0}(x)q_{t_0}^*(x)dx}_{\text{boundary part}} + \tag{27}$$

$$+ \underbrace{\int_{t_0}^{t_1}\int q_t^*(x)\left[\frac{1}{2}\omega(t)\|\nabla s_t(x)\|^2 + \omega(t)\frac{\partial s_t(x)}{\partial t} + s_t(x)\frac{\partial \omega(t)}{\partial t}\right]dx dt}_{\text{middle part}}, \tag{28}$$

which consists of two parts. Estimation of the boundary part involves only sampling from $q_{t_0}^*$ and $q_{t_1}^*$, while the middle part estimate depends on the distribution of time samples, i.e.,

$$\int_{t_0}^{t_1}\underbrace{\frac{p(t)}{p(t)}}_{=1}\int q_t^*(x)\left[\frac{1}{2}\omega(t)\|\nabla s_t(x)\|^2 + \omega(t)\frac{\partial s_t(x)}{\partial t} + s_t(x)\frac{\partial \omega(t)}{\partial t}\right]dx dt \simeq \tag{29}$$

$$\simeq \mathbb{E}_{t\sim p(t)}\mathbb{E}_{x\sim q_t^*(x)}\frac{1}{p(t)}\left[\frac{1}{2}\omega(t)\|\nabla s_t(x)\|^2 + \omega(t)\frac{\partial s_t(x)}{\partial t} + s_t(x)\frac{\partial \omega(t)}{\partial t}\right]. \tag{30}$$

Note that for every choice of $p(t)$ we get an unbiased estimate of the original objective function. Thus, we can design $p(t)$ to reduce the variance of the middle part of the objective. In our experiments, we observed that simply taking $p(t)$ proportionally to the standard deviation of the corresponding integrand significantly reduces the variance, i.e.,

$$p(t) \propto \sqrt{\mathbb{E}_{x \sim q_t}(\zeta_t - \mathbb{E}_{x \sim q_t}\zeta_t)^2}, \quad \zeta_t = \frac{1}{2}\omega(t)\|\nabla s_t(x)\|^2 + \omega(t)\frac{\partial s_t(x)}{\partial t} + s_t(x)\frac{\partial \omega(t)}{\partial t}. \quad (31)$$

We implement sampling from this distribution by aggregating the estimated variances throughout the training with exponential moving average and then follow by linear interpolation between the estimates.

## C "SPARSE DATA" REGIME

We start with the case where the dataset consists only of a single point $x_0 \in \mathbb{R}^d$

$$q_0(x) = \delta(x - x_0), \quad k_t(x_t \mid x) = \mathcal{N}(x_t \mid f_t(x), \sigma_t^2). \quad (32)$$

Then the distribution at time $t$ is

$$q_t(x) = \int dx' \, q_0(x')k_t(x \mid x') = \mathcal{N}(x \mid f_t(x_0), \sigma_t^2). \quad (33)$$

The ground truth vector field $v$ comes from the continuity equation

$$\frac{\partial q_t}{\partial t} = -\langle \nabla, q_t v \rangle \implies \frac{\partial}{\partial t} \log q_t = -\langle \nabla \log q_t, v \rangle - \langle \nabla, v \rangle. \quad (34)$$

For our dynamics, we have

$$\frac{\partial}{\partial t} \log q_t = \frac{\partial}{\partial t}\left[ -\frac{d}{2}\log(2\pi\sigma_t^2) - \frac{1}{2\sigma_t^2}\|x - f_t(x_0)\|^2 \right] \quad (35)$$

$$= -d\frac{\partial}{\partial t}\log \sigma_t + \frac{1}{\sigma_t^2}\|x - f_t(x_0)\|^2 \frac{\partial}{\partial t}\log \sigma_t + \frac{1}{\sigma_t^2}\left\langle x - f_t(x_0), \frac{\partial f_t(x_0)}{\partial t} \right\rangle \quad (36)$$

$$= -d\frac{\partial}{\partial t}\log \sigma_t + \frac{1}{\sigma_t^2}\left\langle x - f_t(x_0), (x - f_t(x_0))\frac{\partial}{\partial t}\log \sigma_t + \frac{\partial f_t(x_0)}{\partial t} \right\rangle; \quad (37)$$

$$\nabla \log q_t = -\frac{1}{\sigma_t^2}(x - f_t(x_0)); \quad (38)$$

$$\frac{\partial}{\partial t}\log q_t = -d\frac{\partial}{\partial t}\log \sigma_t - \left\langle \nabla \log q_t, (x - f_t(x_0))\frac{\partial}{\partial t}\log \sigma_t + \frac{\partial f_t(x_0)}{\partial t} \right\rangle. \quad (39)$$

Matching the corresponding terms in the continuity equation, we get

$$v = (x - f_t(x_0))\frac{\partial}{\partial t}\log \sigma_t + \frac{\partial f_t(x_0)}{\partial t}. \quad (40)$$

For the set of delta-functions, we denote

$$q_0(x) = \sum_i \delta(x - x^i), \quad q_t(x) = \sum_i q_t^i(x), \quad q_t^i(x) = \mathcal{N}(x \mid f_t(x^i), \sigma_t^2). \quad (41)$$

Due to the linearity of the continuity equation w.r.t. $q$, we have

$$\sum_i \frac{\partial q_t^i}{\partial t} = \sum_i \langle \nabla, q_t^i v \rangle \implies \sum_i q_t^i\left(\frac{\partial}{\partial t}\log q_t^i + \langle \nabla \log q_t^i, v \rangle + \langle \nabla, v \rangle\right) = 0. \quad (42)$$

We first solve the equation for $\frac{\partial f_t}{\partial t} = 0$, then for $\frac{\partial}{\partial t}\log \sigma_t = 0$ and join the solutions. For $\frac{\partial f_t}{\partial t} = 0$, we look for the solution in the following form

$$v_\sigma = \frac{A}{\sum_i q_t^i}\sum_i \nabla q_t^i, \quad q_t^i(x) = \mathcal{N}(x \mid f_t^i(x^i), \sigma_t^2). \quad (43)$$

Then we have

$$\langle \nabla, v_\sigma \rangle = \left\langle \nabla \frac{A}{\sum_i q_t^i}, \sum_i \nabla q_t^i \right\rangle + \frac{A}{\sum_i q_t^i} \sum_i \nabla^2 q_t^i \tag{44}$$

$$= -\frac{A}{(\sum_i q_t^i)^2} \left\| \sum_i \nabla q_t^i \right\|^2 + \frac{A}{\sum_i q_t^i} \sum_i q_t^i \left[ \left\| \nabla \log q_t^i \right\|^2 - \frac{d}{\sigma^2} \right], \tag{45}$$

$$\left( \sum_i q_t^i \right) \langle \nabla, v_\sigma \rangle = -\frac{A}{\sum_i q_t^i} \left\| \sum_i \nabla q_t^i \right\|^2 + A \sum_i q_t^i \left[ \left\| \nabla \log q_t^i \right\|^2 - \frac{d}{\sigma^2} \right], \tag{46}$$

and from (42) we have

$$\sum_i q_t^i \left( -d \frac{\partial}{\partial t} \log \sigma_t + \left\langle \nabla \log q_t^i, v_\sigma + \sigma_t^2 \frac{\partial}{\partial t} \log \sigma_t \nabla \log q_t^i \right\rangle + \langle \nabla, v_\sigma \rangle \right) = 0. \tag{47}$$

From these two equations we have

$$\sum_i q_t^i \langle \nabla, v_\sigma \rangle = -\frac{A}{\sum_i q_t^i} \left\| \sum_i \nabla q_t^i \right\|^2 + A \sum_i q_t^i \left[ \left\| \nabla \log q_t^i \right\|^2 - \frac{d}{\sigma^2} \right] = \tag{48}$$

$$= \sum_i q_t^i \left( d \frac{\partial}{\partial t} \log \sigma_t \right) - \frac{A}{\sum_i q_t^i} \left\| \sum_i \nabla q_t^i \right\|^2 - \sigma_t^2 \frac{\partial}{\partial t} \log \sigma_t \sum_i q_t^i \left\| \nabla \log q_t^i \right\|^2. \tag{49}$$

Thus, we have

$$A = -\sigma_t^2 \frac{\partial}{\partial t} \log \sigma_t. \tag{50}$$

For $\frac{\partial}{\partial t} \log \sigma_t = 0$, we simply check that the solution is

$$v_f = \frac{1}{\sum_i q_t^i} \sum_i q_t^i \frac{\partial f_t(x^i)}{\partial t}. \tag{51}$$

Indeed, the continuity equation turns into

$$\sum_i q_t^i \left( \left\langle \nabla \log q_t^i, v_f - \frac{\partial f_t(x^i)}{\partial t} \right\rangle + \langle \nabla, v_f \rangle \right) = 0. \tag{52}$$

From the solution and the continuity equation we write $\sum_i q_i \langle \nabla, v_f \rangle$ in two different ways.

$$\sum_i q_t^i \langle \nabla, v_f \rangle = -\frac{1}{\sum_i q_t^i} \left\langle \sum_i \nabla q_t^i, \sum_i q_t^i \frac{\partial f_t(x^i)}{\partial t} \right\rangle + \sum_i \left\langle \nabla q_t^i, \frac{\partial f_t(x^i)}{\partial t} \right\rangle \tag{53}$$

$$= -\left\langle \sum_i \nabla q_t^i, v_f \right\rangle + \sum_i \left\langle \nabla q_t^i, \frac{\partial f_t(x^i)}{\partial t} \right\rangle \tag{54}$$

Thus, we see that (51) is indeed a solution.

Finally, unifying $v_\sigma$ and $v_f$, we have the full solution

$$v = -\left( \frac{\partial}{\partial t} \log \sigma_t \right) \frac{\sigma_t^2}{\sum_i q_t^i} \sum_i \nabla q_t^i + \frac{1}{\sum_i q_t^i} \sum_i q_t^i \frac{\partial f_t(x^i)}{\partial t}, \quad q_t^i(x) = \mathcal{N}(x \mid f_t(x^i), \sigma_t^2), \tag{55}$$

$$v = \frac{1}{\sum_i q_t^i} \sum_i q_t^i \left[ (x - f_t(x^i)) \frac{\partial}{\partial t} \log \sigma_t + \frac{\partial f_t(x^i)}{\partial t} \right] \tag{56}$$

input             colorizations

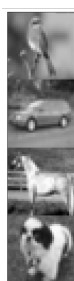 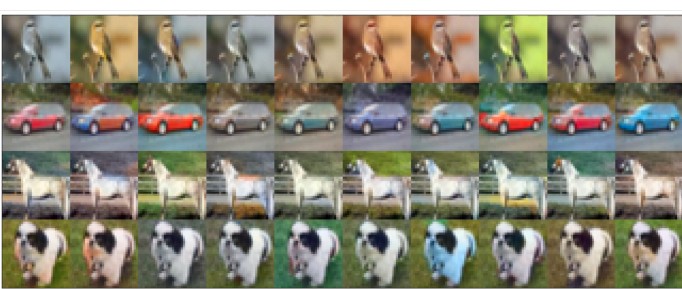

Figure 6: Illustration that Action Matching can learn one to many relations using low variance noise added to the image. Here, we sample different colorizations starting from the same grayscale input adding different samples of noise.

# D    IMPLEMENTATION DETAILS

## D.1    DETAILS OF ACTION MATCHING GENERATIVE MODELS

For the architecture of the neural network parameterizing $s_t$, we follow (Salimans & Ho, 2021). In more details, we parameterize $s_t(x)$ as $\|\text{unet}(t, x) - x\|^2$, where $\text{unet}(t, x)$ is the U-net architecture (Ronneberger et al., 2015). For the U-net architecture, we follow (Song et al., 2020b) with the only difference is that we set the channel multiplier parameter to $64$ instead of $128$, thus, narrowing down the architecture. We have to narrow down the architecture since Action Matching requires taking the derivative w.r.t. the inputs at each iteration, which is a downside compared to Denoising Score Matching. Otherwise the training of one model takes a week on 4 gpus. We consider the same U-net architecture for the baseline to parameterize $\nabla \log q_t$.

For diffusion, we take VP-SDE from (Song et al., 2020b), which corresponds to $\alpha_t = \exp(-\frac{1}{2} \int \beta(s) ds)$ and $\sigma_t = \sqrt{1 - \exp(-\int \beta(s) ds)}$, where $\beta(s) = 0.1 + 19.9t$. For other tasks we take $\sigma_t = t$ and $\alpha_t = 1 - t$. All images are normalized to the interval $[-1, 1]$. For image generation on the torus, we first normalize the data such that every pixel is in $[0.25, 0.75]$. Thus we make sure that the shortest distance between the lowest and the largest pixel values is maximal on the circle $[0, 1]$.

Although Action Matching learns deterministic mappings, it is possible to learn one-to-many mappings by adding small amount of noise to the data. For example, each row of Fig. 6 shows that Action Matching has learned to generate different colorizations from a single grayscale CIFAR-10 image, using different noise samples added to the grayscale image in Eq. (14).

## D.2    DETAILS ON THE SCHRÖDINGER EQUATION SIMULATION

For the initial state of the dynamics

$$i \frac{\partial}{\partial t} \psi(x, t) = -\frac{1}{\|x\|} \psi(x, t) - \frac{1}{2} \nabla^2 \psi(x, t), \tag{57}$$

we take the following wavefunction

$$\psi(x, t = 0) \propto \psi_{32-1}(x) + \psi_{210}(x), \quad \text{and} \quad q^*_{t=0}(x) = |\psi(x, t = 0)|^2, \tag{58}$$

where $n, l, m$ are quantum numbers and $\psi_{nlm}$ is the eigenstate of the corresponding Hamiltonian (see Griffiths & Schroeter (2018)). For all the details on sampling and the exact formulas for the initial state, we refer the reader to the code github.com/action-matching. We evolve the initial state for $T = 14 \cdot 10^3$ time units in the system $\hbar = 1, m_e = 1, e = 1, \varepsilon_0 = 1$ collecting the dataset of samples from $q^*_t$. For the time discretization, we take $10^3$ steps; hence, we sample every $14$ time units.

To evaluate each method, we collect all the generated samples from the distributions $q_t$, $t \in [0, T]$ comparing them with the samples from the training data. For the distance metric, we measure the MMD distance (Gretton et al., 2012) between the generated samples and the training data at $10$ different timesteps $t = \frac{k}{10} T$, $k = 1, \dots, 10$ and average the distance over the timesteps. For the

Annealed Langevin Dynamics, we set the number of intermediate steps for $M = 5$, and select the step size $dt$ by minimizing MMD using the exact scores $\nabla \log q_t(x)$.

For all methods, we use the same architecture, which is a multilayer perceptron with 5 layers 256 hidden units each. The architecture $h(t, x)$ takes $x \in \mathbb{R}^3$ and $t \in \mathbb{R}$ and outputs 3-d vector, i.e. $h(t, x) : \mathbb{R} \times \mathbb{R}^3 \to \mathbb{R}^3$. For the score-based models it already defines the score, while for action matching we use $s_t(x) = \|h(t, x) - x\|^2$ as the model and the vector field is defined as $\nabla s_t(x)$.

---

**Algorithm 3** Annealed Langevin Dynamics

---

**Require:** score model $s_t(x)$, step size $dt$, number of intermediate steps $M$
**Require:** initial samples $x_0^i \in \mathbb{R}^d$
  **for** time steps $t \in (0, T]$ **do**
    set the target distribution $q_t$, such that $s_t(x) \approx \nabla \log q_t(x)$
    **for** intermediate steps $j \in 1, \ldots, M$ **do**
      $\varepsilon^i \sim \mathcal{N}(0, \mathbf{1})$
      $x_t^i = x_t^i + \frac{dt}{2} s_t(x_t^i) + \sqrt{dt} \cdot \varepsilon^i$
    **end for**
    save samples $x_t^i$
  **end for**
  **return** samples $\{x_t^i\}_{t=0}^T$

---

## E IMAGE EXAMPLES FOR ACTION MATCHING

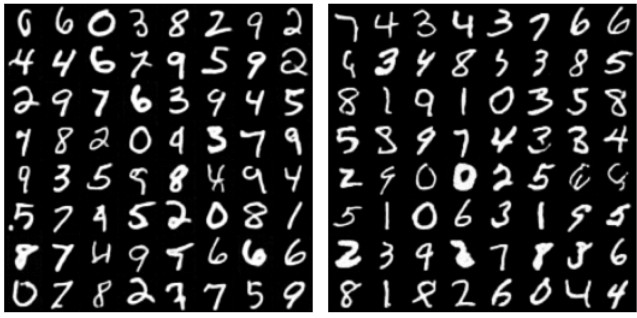

Figure 7: Action Matching on MNIST diffusion (on the left), torus (on the right).

diffusion

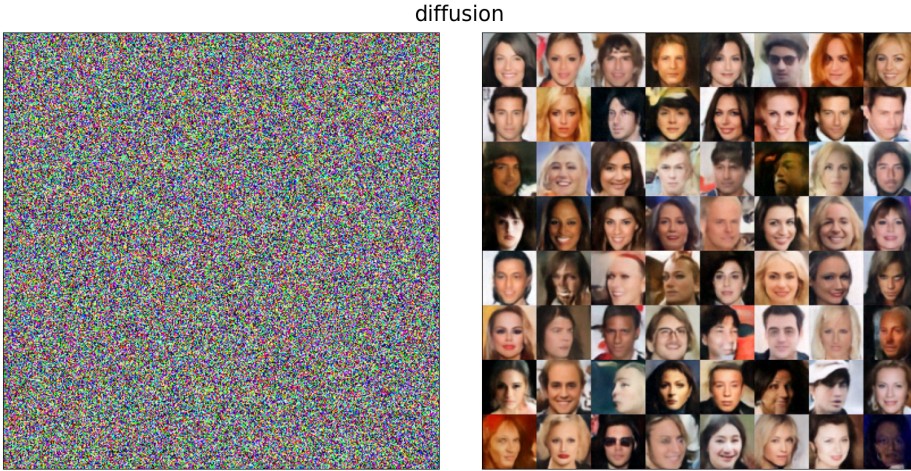

Figure 8: Action Matching on CelebA for diffusion.

superres

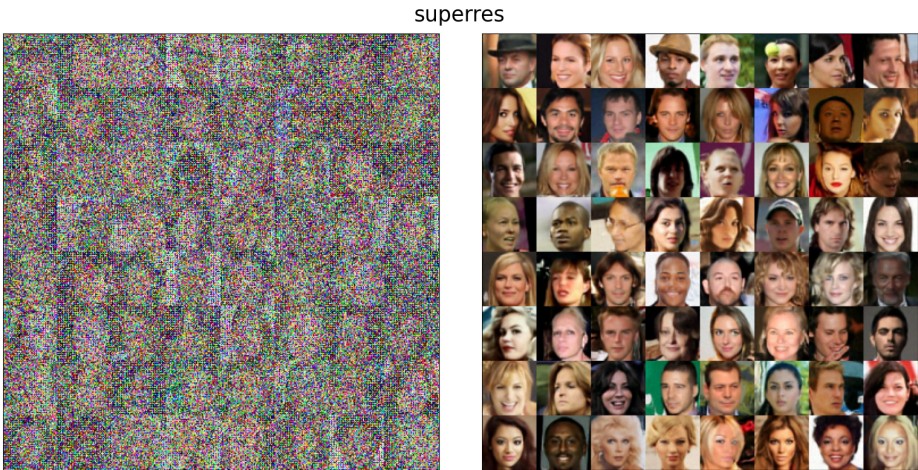

Figure 9: Action Matching on CelebA for superres.

inpaint

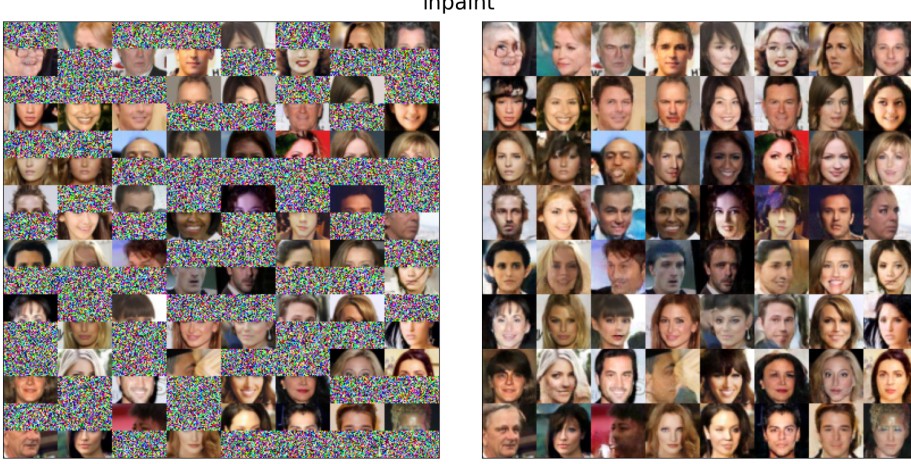

Figure 10: Action Matching on CelebA for inpaint.

torus

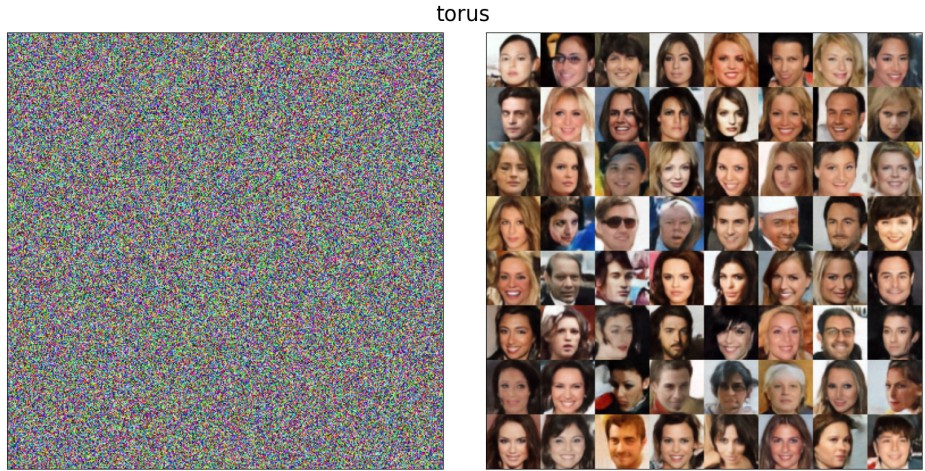

Figure 11: Action Matching on CelebA for torus.

diffusion

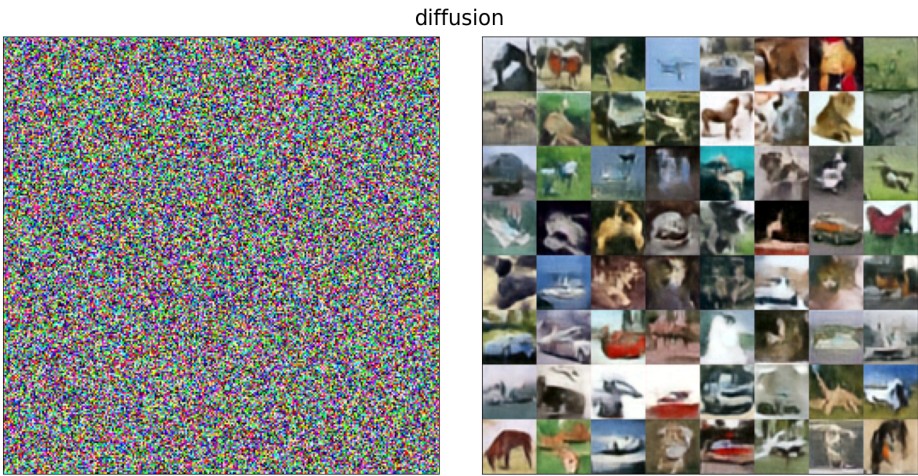

Figure 12: Action Matching on CIFAR-10 for diffusion.

superres

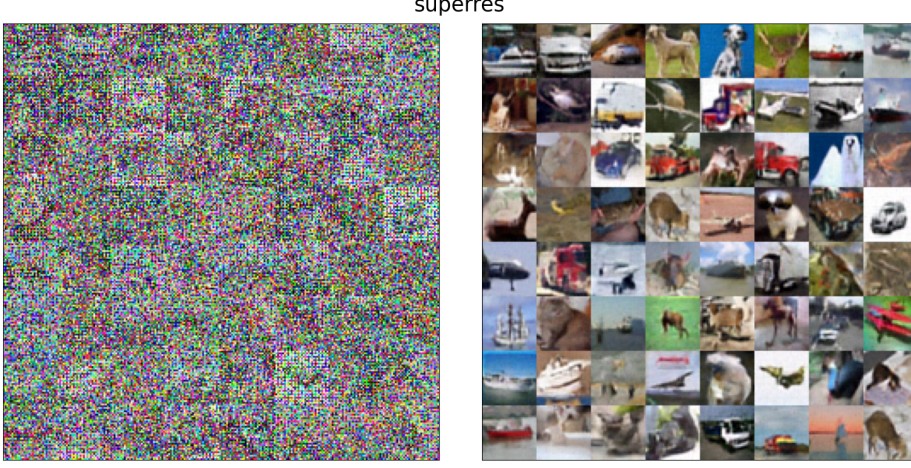

Figure 13: Action Matching on CIFAR-10 for super-resolution.

color

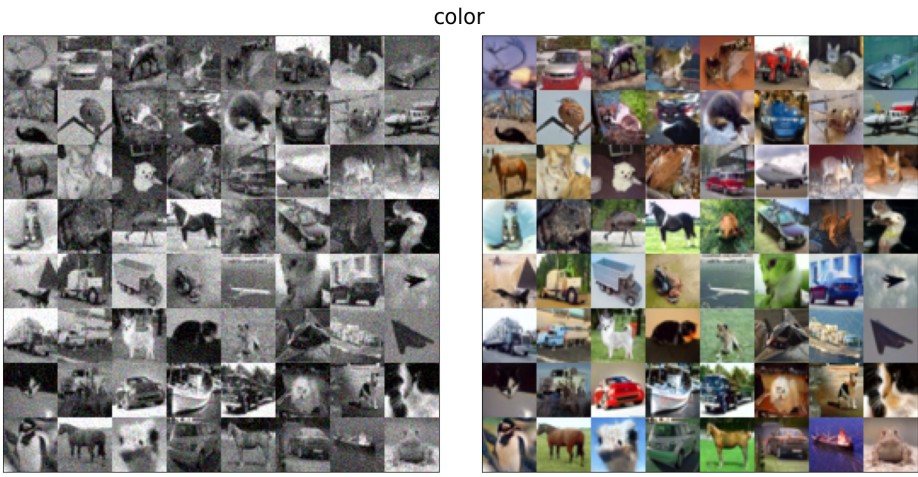

Figure 14: Action Matching on CIFAR-10 for colorization.

torus

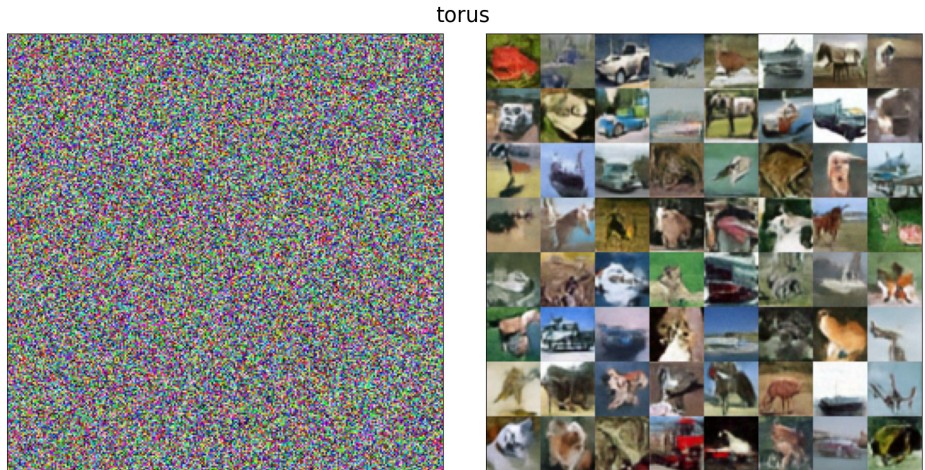

Figure 15: Action Matching on CIFAR-10 for torus.

