# OpenReview forum: "Action Matching: A Variational Method for Learning Stochastic Dynamics from Samples"
_ICLR.cc/2023/Conference — Submitted to ICLR 2023_

### Official Review · Reviewer_Sbd8 · 2022-10-24

**Confidence:** 3
**Correctness:** 3
**Technical Novelty And Significance:** 2
**Empirical Novelty And Significance:** 3
**Recommendation:** 5

**Clarity, Quality, Novelty And Reproducibility:**

The presentation was clear in total but I found the above unclear points. The quality may be good in total because of the above contributions and strengths. The experiments may not be reproduced because they did not provide the code.


**Strength And Weaknesses:**

Strength:
* The idea of the above first contribution is interesting and may have a novelty.
* The experimental results clearly show the superiority of the proposed approach.

Weakness:
* The presentations (in particular the first half of the main text) and figures were unclear to me, which is described as the specific comments.
* It may be a small weakness: in my understanding, the generative model of the proposed approach needs the definition of dynamics, but the previous action-matching methods also require it.

Specific comments
* In the Introduction, q_t and “score matching” were not defined and then Fig. 1 was meaningless.
* In my understanding, this method can apply to the dynamics which can be expressed as curl free vector fields. Is this correct? Are there any other dynamics than you analyzed?
* Fig. 3: what are the two types of pictures for Score/Action matching?
* Fig. 4: If they are generated, the information of input and the dynamics may be necessary for understanding. Moreover, MINST was not used in the main text.
* FID may be not defined.


**Summary Of The Paper:**

The authors proposed a method that enables us to learn a broader family of stochastic dynamics by directly learning an underlying mechanism to move samples in time without modeling the distributions at each time-step. The contributions are as follows:

1. Action Matching relies only on samples and does not require any knowledge of the underlying stochastic dynamics by learning only the dynamics, which is based on a mathematically well-defined formulation and useful when a complicated distribution has simple dynamics.
2. They discussed Action Matching as a variational framework for learning these dynamics
3. For applications of Action Matching, they performed computer vision tasks with several stochastic equations such as generative modeling, super-resolution, colorization, and inpainting.



**Summary Of The Review:**

Based on the comment above, I consider that the weakness of this paper outperformed the strength at this stage, then I cannot give a higher rating.

---

> ### Author Response · Authors · 2022-11-12
> **Response to Reviewer Sbd8**
>
> Thank you for your review and your suggestion for improving the presentation! We have added another set of experiments in Section 5.2, where we show that Action Matching can accurately learn the dynamics of quantum systems (see Section 5.2 in the paper)
> In what follows, we address your specific comments.
>
> > “It may be a small weakness: in my understanding, the generative model of the proposed approach needs the definition of dynamics, but the previous action-matching methods also require it.”
>
> We would like to clarify that Action Matching does **not** require the definition of dynamics and only requires access to samples from the dynamics, which can be given as training data. In none of our experiments, we used the definition of dynamics for learning Action Matching models. For example, in the new experiment, we learn the dynamics of a quantum system purely from training data without any knowledge about the underlying dynamics (Schrödinger equation). In contrast, previous methods, as you pointed out, do require access to both samples and the analytic model of the dynamics.
>
> > “In the Introduction, q_t and “score matching” were not defined and then Fig. 1 was meaningless.”
>
> We have now added the mention of Score Matching to the introduction, and $q_t$ has already been introduced in the second paragraph of the introduction.
>
> > “In my understanding, this method can apply to the dynamics which can be expressed as curl free vector fields. Is this correct? Are there any other dynamics than you analyzed?”
>
> According to Theorem 1, almost any dynamics of interest in physical science (including diffusions) can be expressed by the curl-free vector field. This means that even if the original dynamics was produced by non curl-free vector field in the state space, we still could learn a curl-free vector field that respects all the marginal distributions $q_t$ of the original dynamics. The goal of Section 2 was to discuss this and justify that the curl-free vector fields are by no means a restricting assumption.
> The intuition behind this fact is that any underlying vector field $v_t$ can be decomposed as $v_t=\nabla s_t +u_t$, where the first term is curl-free $\nabla s_t$, and the second term $u_t$ is divergence-free, meaning $\nabla \cdot (q_t u_t)=0$. So if $v_t$ satisfies the continuity equation, the curl-free term $\nabla s_t$ must also satisfy the continuity equation because
>
> $\frac{\partial}{\partial t}q_t = -\nabla \cdot (q_t v_t) =-\nabla \cdot (q_t(\nabla s_t + u_t))= -\nabla \cdot (q_t \nabla s_t)\,$
>
> This is also true if the underlying dynamics is stochastic. So without loss of generality, we only consider the curl-free vectors.
>
> > “Fig. 3: what are the two types of pictures for Score/Action matching?”
>
> As we describe in paragraph 4 of Section 5, we concatenate the input with the conditional information (grayscale image in this case). We have added the corresponding description to the figure’s caption.
>
> > “Fig. 4: If they are generated, the information of input and the dynamics may be necessary for understanding. Moreover, MINST was not used in the main text.”
>
> We have described all the dynamics in Section 4.1, and its hyperparameters in Appendix D. We have added MNIST mention to the main text. We will provide more details on hyperparameters and inputs in our code.
>
> > “FID may be not defined.”
>
> We have added the corresponding reference to the main text.
>
> > “The experiments may not be reproduced because they did not provide the code.”
>
> We have open-sourced our code for reproducing the experiments of Section 5.2 in our [github repository](https://github.com/action-matching/action-matching#schr%C3%B6dinger-equation-simulation). The code for the remaining experiments will be open-sourced with the next version of the paper.

---

> > ### Comment · Reviewer_Sbd8 · 2022-12-04
> > **Thank you for the response**
> >
> > Thank you for the response and sorry for some misunderstandings.
> > I understood the contents of the rebuttal, but at this stage, I did not raise the score (due to the lack of clear presentation) .

---

### Official Review · Reviewer_Ln22 · 2022-10-24

**Confidence:** 4
**Correctness:** 3
**Technical Novelty And Significance:** 2
**Empirical Novelty And Significance:** 2
**Recommendation:** 5

**Clarity, Quality, Novelty And Reproducibility:**

The paper is well written. However, based on the descriptions of the paper, reproducing the results is not trivial.

**Strength And Weaknesses:**

Strength:

Using action matching to avoid modeling the distributions at each time-step is interesting. Experimental results are able to demonstrate the effect of the proposed method.

Weaknesses:

Action matching is based on the diffusion model. However, it is not clear whether the proposed action matching performs better or not.

In addition, it lacks comparisons the with the related methods. Only showing the results of the proposed method is not enough.

It is not clear whether the proposed action matching has a better convergence property or not.

**Summary Of The Paper:**

This paper an action matching method to learn broader family of stochastic dynamics. Instead of modeling the distributions at each time-step, the action matching directly learns an underlying mechanism to move samples in time. Experimental results show that the proposed method can be applied to image super-resolution, colorization, and so on.


**Summary Of The Review:**

Lacking analysis and comparisons with the related methods.

---

> ### Author Response · Authors · 2022-11-12
> **Response to Reviewer Ln22**
>
> Thank you for your feedback! In what follows, we answer the concerns raised and clarify important details. We also add a new experiment to the paper where we show that Action Matching restores the dynamics of quantum systems (see Section 5.2 in the paper). We use this experiment to answer some of your concerns.
>
> > “Action matching is based on the diffusion model.”
>
> We would like to clarify that Action Matching is **not** based on diffusion. The diffusion process is one of many possible stochastic dynamics (distributional path on the Wasserstein manifold). Using Theorem 1, we can learn any distributional path (absolutely-continuous curve on the Wasserstein manifold). To give an example, we provide the experiment where we learn the dynamics of a quantum system (according to the Schrödinger equation), which is not a diffusion and cannot be simulated by the diffusion. To be precise, the only two diffusion processes we consider in the paper are “diffusion” and “torus” in Fig. 2. Note that the “torus” dynamics defines the diffusion process on a manifold and thus does not allow for a straightforward application of the Denoising Score Matching.
>
> > “In addition, it lacks comparisons the with the related methods. Only showing the results of the proposed method is not enough.”
>
> In Section 5.1, we compare the proposed method to methods from papers [1,2] (see Table 1 and Fig 4). In our new experiments in Section 5.2, we compare Action Matching against the methods from papers [3,4,5] (see Table 2 and Fig 5).
>
> [1] Song Y, Sohl-Dickstein J, Kingma DP, Kumar A, Ermon S, Poole B. Score-based generative modeling through stochastic differential equations. arXiv preprint arXiv:2011.13456. 2020 Nov 26.\
> [2] Saharia C, Chan W, Chang H, Lee C, Ho J, Salimans T, Fleet D, Norouzi M. Palette: Image-to-image diffusion models. In ACM SIGGRAPH 2022 Conference Proceedings 2022 Jul 27 (pp. 1-10).\
> [3] Song Y, Ermon S. Generative modeling by estimating gradients of the data distribution. Advances in Neural Information Processing Systems. 2019;32.\
> [4] Hyvärinen A, Dayan P. Estimation of non-normalized statistical models by score matching. Journal of Machine Learning Research. 2005 Apr 1;6(4).\
> [5] Song Y, Garg S, Shi J, Ermon S. Sliced score matching: A scalable approach to density and score estimation. In Uncertainty in Artificial Intelligence 2020 Aug 6 (pp. 574-584). PMLR.
>
> >“It is not clear whether the proposed action matching has a better convergence property or not.”
>
> For generative modeling, the proposed method may demonstrate better convergence properties due to the flexibility of the formulation. We demonstrate it on the colorization task (see Fig 4). However, we would like to point out that the convergence properties of Action Matching were only a side benefit of our method and by no means our main contribution. As we discussed above, many dynamics of interest are not diffusion, and diffusions simply can be used to model them. For example, as we demonstrated by simulating the Schrodinger equation, when it’s impossible to define the analytic model of the dynamics, Action Matching demonstrates superior performance compared to all other baselines.
>
> > “based on the descriptions of the paper, reproducing the results is not trivial.”
>
> We have open-sourced our code for reproducing the experiments of Section 5.2 in our [github repository](https://github.com/action-matching/action-matching#schr%C3%B6dinger-equation-simulation). The code for the remaining experiments will be open-sourced with the next version of the paper.

---

### Official Review · Reviewer_dWGD · 2022-10-24

**Confidence:** 3
**Clarity, Quality, Novelty And Reproducibility:** The writing is clear and easy to foll…
**Correctness:** 4
**Technical Novelty And Significance:** 3
**Empirical Novelty And Significance:** 3
**Recommendation:** 6

**Strength And Weaknesses:**

Strengths:
+ The proposed method is theoretically sound and general. It can potentially have a wide range of impact beyond image generation.
+ Multiple vision tasks are evaluated, including diffusion, colorization, and impainting. Furthermore, results show consistent improvement on multiple image datasets. I find the results convincing.

Weaknesses:
- While the proposed method is general, the experiments are limited to low-resolution image generation tasks. Experiments on tasks beyond image generation (e.g. natural sciences ones), even just toy examples, can be interesting.


**Summary Of The Paper:**

In this paper, the authors show that many continuous dynamics can be represented as a continuous vector field and propose a method called action matching that can learn the vector field. One key idea is that instead of learning the distributions themselves, the method aims to learn only the dynamics. This can be effective when dynamics are simple but distributions are complex. On multiple image generation tasks, the method outperforms score matching by a clear margin.




**Summary Of The Review:**

Overall the study presented in the paper is sound and promising. Both the theoretical and empirical results are convincing.

---

> ### Author Response · Authors · 2022-11-12
> **Response to Reviewer dWGD**
>
> Thank you for your review! We are glad that you found Action Matching “novel and original” and appreciated its “wide range of impact beyond image generation.”
>
> > “Experiments on tasks beyond image generation (e.g., natural sciences ones), even just toy examples, can be interesting.”
>
> As per your suggestion, we added an experiment motivated by natural sciences. We demonstrate that Action Matching is able to simulate the dynamics of a quantum system precisely. Namely, we consider the evolution of an excited state of the hydrogen atom in the corresponding potential. In Section 5.2, we demonstrate that Action Matching outperforms the previous methods, and we provide code and visualization in our [github repository](https://github.com/action-matching/action-matching#schr%C3%B6dinger-equation-simulation), showing that the Action Matching dynamics matches the ground-truth dynamics described by the Schrödinger equation.
>
> > “On multiple image generation tasks, the method outperforms score matching by a clear margin.”
>
> There is a misunderstanding about the performance of the proposed method. In Table 1, Action Matching performs worse than Score Matching. This is expected since Denoising Score Matching (which we use for experiments) heavily relies on the analytic form of the noising process, thus having more information about the problem. In other words, Denoising Score Matching should be viewed as an upper-bound in performance that action matching tries to match with less information. However, for instance, Denoising Score Matching is not able to learn the noising process on a manifold (‘torus’ example in Table 1). At the same time, Action Matching allows for performance improvement due to the flexibility of the formulation (see the colorization example in Fig 4). Note that many practical distributional paths, such as the quantum mechanics experiment, are not diffusion, and other methods, such as Denoising Score Matching, simply cannot be applied to them. However, as we demonstrated in the quantum mechanics experiment of the paper, Action Matching can learn these dynamics and outperform other baselines simulating the Schrödinger equation.

---

### Official Review · Reviewer_iYDX · 2022-10-25

**Confidence:** 3
**Correctness:** 3
**Technical Novelty And Significance:** 2
**Empirical Novelty And Significance:** 1
**Recommendation:** 3

**Clarity, Quality, Novelty And Reproducibility:**

I did not enjoy the writing in this paper. There is a lot of meandering and discussion that is neither rigorous nor relevant to the target audience of this conference. E.g.
"For example, in Quantum Mechanics, the act of measurement at a given point collapses the wave function which prevents us from obtaining further samples along that trajectory." is a particularly egregious example, but others throughout.

The most important parts of training procedure and loss are glossed over and presented in a roundabout way, which is not at all clear (e.g., the "analogy with VI and generative modeling" paragraph was very hand-wavy not a good use of space.).

The citations section is also way too sparse.

**Strength And Weaknesses:**

I don't think the method proposed in this paper is new.
It seems to be the same as the idea of learning "time scores", e.g., change in distribution through time.
https://arxiv.org/pdf/2111.11010.pdf

Furthermore, the paper claims that Action Matching can be preferred over score matching since score matching can only handle the forward process of noising, and not of more general transformations like decolorization. I don't think this is true, e.g., this paper below.
Their experimental results are much better than the ones presented here.
https://arxiv.org/pdf/2201.11793.pdf

**Summary Of The Paper:**

This paper proposes a method called Action Matching for learning generative models. In particular, this technique falls along the lines of score-based / diffusion probabilistic models, by learning about the 'derivative' of the probability distribution rather than directly modeling the density.

The paper claims that the main difference between action matching and score matching is that action matching learns about the derivative along the time axis. The argument is that a distribution can be complicated to model for each time slice, but actually has very little change w.r.t. time, so modeling this change w.r.t. time can make the learning problem easier.

**Summary Of The Review:**

The paper proposes Action Matching to learn the change in distribution through time. The paper claims that Action Matching can handle more general forward processes (decolorization, masking...).

However, it appears that the idea of learning "time scores" is not new, and previous diffusion based methods can already handle these more general forward processes. Furthermore, the writing is not clear and concise, and the experimental results are weak.

---

> ### Author Response · Authors · 2022-11-12
> **Response to Reviewer iYDX (Part 1)**
>
> Thank you for your time spent on the review! We believe that there is a crucial misunderstanding of the proposed method, and in what follows, we are addressing this misunderstanding and the concerns raised.
>
> > “The paper claims that the main difference between action matching and score matching is that action matching learns about the derivative along the time axis.” “I don't think the method proposed in this paper is new. It seems to be the same as the idea of learning "time scores", e.g., change in distribution through time [1]”
>
> We believe that there is a significant misunderstanding in the connection of our work with the time-score paper [1]. The problem they are trying to solve, their objective, the conceptual meaning of $s_t(x)$, and its optimal value are different from ours. The time-score paper solves the problem of learning density-ratios by learning derivatives of the log probabilities along the time axis. However, Action Matching learns to sample from the distributional path by learning a velocity vector field that describes how to move the particle in the *state space*, and does not learn the changes in the *density function space* as the time-score paper does. More precisely, we learn the vector field $v_t(x)$ that moves samples respecting the continuity equation $\partial_t q_t(x) = - \nabla_x \cdot (q_t(x) v_t(x))$. This equation is fundamental since it allows us to respect all the marginal densities for the continuous-time processes. Furthermore, Theorem 1 allows us to parameterize the vector field as $v_t(x) = \nabla_x s_t(x)$, where $s_t(x): \mathbb{R}^d \to \mathbb{R}$ is a scalar-valued function defined on the state space. Learning such a function enables us to sample exactly from the dynamics without any approximations or convergence at infinity (as in MCMC). Note that simply learning the “time scores” does not allow for this.
>
> [1] Choi K, Meng C, Song Y, Ermon S. Density ratio estimation via infinitesimal classification. In International Conference on Artificial Intelligence and Statistics 2022 May 3 (pp. 2552-2573). PMLR.
>
> > “Furthermore, the paper claims that Action Matching can be preferred over score matching since score matching can only handle the forward process of noising, and not of more general transformations like decolorization. I don't think this is true, e.g., this paper below. Their experimental results are much better than the ones presented here [1]”
>
> The corruption process considered in the DDRM paper [1] is a one-step procedure of $y=Hx+z$ rather than a forward continuous-time process of Action Matching. The denoising process that they consider simply uses a pre-trained discrete-time diffusion model to sample from the posterior of $p(y|x)$. This is in contrast to our work, where the backward denoising process directly inverts the forward noising process. In other words, in the DDRM paper, there is no ground truth dynamics to restore. In our paper, there is no difference between the forward and the backward processes in that we can simulate the *same* ODE $\dot{x} = \nabla s_t(x)$ forward or backward in time to sample from forward or backward processes respectively. That is, the backward (generation) process has the same marginals as the forward (corruption) process in every moment of (continuous) time. More generally, we never claimed that our paper achieves the best colorization results, and the purpose of these experiments is to back up the claim of our paper, which is *Action Matching can trace any high-dimensional distribution path*, that in this case, happened to be a particular path corresponding to colorization. To the best of our knowledge, our work is the first work that proposes a tractable solution to the distribution path matching problem from samples.
>
> [1] Kawar B, Elad M, Ermon S, Song J. Denoising diffusion restoration models. arXiv preprint arXiv:2201.11793. 2022 Jan 27.
>
> > “I did not enjoy the writing in this paper. There is a lot of meandering and discussion that is neither rigorous nor relevant to the target audience of this conference. E.g. "For example, in Quantum Mechanics, the act of measurement at a given point collapses the wave function which prevents us from obtaining further samples along that trajectory." is a particularly egregious example, but others throughout.”
>
> Applications of machine learning to physics have always been a relevant topic for the ML community. We have added a new experiment to the paper, demonstrating that Action Matching accurately learns the dynamics of a quantum system (see Section 5.2). We describe all the prerequisites in an accessible manner and provide the necessary references and the code reproducing the experiments. We believe that this experiment explains our interest in applying Action Matching to quantum systems and explains this particular in the introduction.

---

> > ### Author Response · Authors · 2022-11-12
> > **Response to Reviewer iYDX (Part 2)**
> >
> > > “The most important parts of training procedure and loss are glossed over and presented in a roundabout way, which is not at all clear ”
> >
> > The step-by-step pseudocode for the training procedure and the loss function is provided in Algorithm 1 of the paper. The loss function is explicitly stated in Eq. 8. Sections 2 and 3 are dedicated to describing the loss function and providing intuition for it. More specifically, we introduce the loss function by first using Theorem 1, which explains that a large family of dynamics can be simulated via the continuity equation, and then, in Proposition 1, presenting how the vector field satisfying the continuity equation can be learned.
> >
> > > “the "analogy with VI and generative modeling" paragraph was very hand-wavy not a good use of space.”
> >
> > In the paper, we do our best to present the method from different perspectives and provide intuition on different levels. A large part of the community relies on the intuition provided by the classic works on Variational Auto-Encoders, which motivated us to draw such a connection. However, as per your suggestion, we decided to remove this piece in favor of the new experiment.

---

### Author Response · Authors · 2022-11-12
**Common Response**

Thank you for your reviews and the time spent!

**Action Matching is much more general than Diffusion**: We would like to point out that Action Matching can learn dynamics that are much more general than diffusion. Indeed, as described by Theorem 1, Action Matching can learn any continuous dynamics, while diffusions constitute only a small portion of the space of all dynamics. Most of the dynamics that we study in this paper are not diffusions. For example, one cannot apply denoising score-matching to learn the non-diffusion dynamics of our super-resolution or in-painting experiment in Eq. 13, 14 (In Table 1, we have used a separate diffusion dynamics for super-resolution and in-painting, but these dynamics do not follow the non-diffusion dynamics of Eq. 13, 14). Unlike Score Matching, Action Matching can be used to learn dynamics on a manifold like the torus examples. Finally, the dynamics of quantum systems is not a diffusion, it is described by the Schrödinger equation, but Action Matching can still be used to learn it as described in our new experiment.

**New Quantum Mechanics Experiment** : Based on your feedback, we perform a new experiment in Section 5.2, in which we use Action Matching to learn the dynamics of a quantum system. We choose this experiment since quantum systems are a natural application of Action Matching, as they involve the evolution of position/momentum densities over time, where the trajectories of the samples cannot be observed even in principle. More precisely, purely from the training samples, we can use Action Matching to *accurately* learn the evolution of the wave function according to the Schrödinger equation in the hydrogen atom potential, whereas other baselines fail to learn this dynamics. We refer you to Section 5.2 for more details and comparison. Furthermore, we have provided the code and the animation of the Action Matching dynamics and other baselines of this experiment in our [github repository](https://github.com/action-matching/action-matching), which shows that Action Matching can qualitatively match the ground-truth dynamics of the Schrödinger equation. This experiment demonstrates that Action Matching has a wide range of applications for inferring the underlying laws governing the evolution of physical systems (in addition to ML applications).

---

### Author Response · Authors · 2022-12-07
**Pedagogical Implementation of Action Matching in JAX**

In order to address the concerns of the reviewers regarding the clarity of “training procedure and loss” and the reproducibility of the results, we have re-implemented Action Matching in JAX, and have provided a clear pedagogical example in this [Jupyter Notebook](https://colab.research.google.com/drive/1UP8ESCSflfulcWMuMiC3_TAYHc-Ny1r3?usp=sharing). The code is short and the main point of the paper is compressed into a single 10-line function (```am_loss```), highlighting the practicality of Action Matching for learning many different dynamics that are more general than diffusion, purely from samples.

---

### Decision · Program_Chairs · 2023-01-20

**Decision:**

Reject

**Justification For Why Not Higher Score:**

This is clearly a rejection from my point of view. I did try to initiate a discussion with the reviewers and sent several emails to them but there was no response. Based on my own reading of the paper and the reviews, I find that two aspects are clearly lacking: discussion of the overlap with prior work and limited experimental results (which I believe is an important aspect as the paper is more on the empirical side).

**Justification For Why Not Lower Score:**

N/A

**Metareview: Summary, Strengths And Weaknesses:**

This paper proposes a method they name Action Matching for modeling stochastic dynamics of generative models. This technique falls along the lines of score-based / diffusion probabilistic models, by learning about the 'derivative' of the probability distribution rather than directly modeling the density.

The reviewers were unanimous in rejecting the paper (the average score of 4.75 is clearly below the acceptance bar). Despite some of the reviews having some incorrect statements (as pointed out by the authors), there are two important points that were raised to which I largely agree: 1) a lack of discussion regarding the overlap with prior work and 2) limited experimental results (which is an important aspect as the paper is more empirical). The reviewers did not change their opinion during the discussion period (although I note that not all reviewers participated in the discussion).

Overall, I think there are some important problems to be fixed and I am therefore not able to recommend acceptance at this time.

**Summary Of Ac-Reviewer Meeting:**

One reviewer and I initiated a discussion in which no other reviewers participated (also sent emails to the reviewers). I, therefore, took a decision based on my own reading of the paper and reviews (which I believe to clearly lean toward rejection).